# Fine-Grained Neural Network Explanation by Identifying Input Features with Predictive Information

**Yang Zhang** [*]
Technical University of Munich
ya.zhang@tum.de

**Ashkan Khakzar** [*]
Technical University of Munich
ashkan.khakzar@tum.de

**Yawei Li**
LMU University Hospital [◇], Munich
yawei.li@med.uni-muenchen.de

**Azade Farshad**
Technical University of Munich
azade.farshad@tum.de

**Seong Tae Kim** [†]
Kyung Hee University, South Korea
st.kim@khu.ac.kr

**Nassir Navab**
Technical University of Munich
nassir.navab@tum.de

## Abstract

One principal approach for illuminating a black-box neural network is feature attribution, i.e. identifying the importance of input features for the network's prediction. The predictive information of features is recently proposed as a proxy for the measure of their importance. So far, the predictive information is only identified for *latent* features by placing an information bottleneck within the network. We propose a method to identify features with predictive information in the *input* domain. The method results in fine-grained identification of input features' information and is agnostic to network architecture. The core idea of our method is leveraging a bottleneck on the input that only lets input features associated with predictive latent features pass through. We compare our method with several feature attribution methods using mainstream feature attribution evaluation experiments. The code [1] is publicly available.

## 1 Introduction

Feature attribution – identifying the contribution of input features to the output of the neural network function – is one principal approach for explaining the predictions of black-box neural networks. In recent years, a plethora of feature attribution methods is proposed. The solutions range from axiomatic methods [1, 2, 3] derived from game theory [4] to contribution backpropagation methods [5, 6, 7, 8, 9]. However, given a specific neural network and an input-output pair, existing feature attribution methods show dissimilar results. In order to evaluate which method correctly explains the prediction, the literature proposes several attribution evaluation experiments [10, 11, 12, 9]. The attribution evaluation experiments reveal that methods that look visually interpretable to humans or even methods with solid theoretical grounding are not identifying contributing input features [13, 12, 14]. The divergent results of different attribution methods and the insights from evaluation experiments show that the feature attribution problem remains unsolved. Though there is no silver

---

[*]denotes equal contribution [◇]Department of Dermatology and Allergology [†]corresponding author
[1]https://github.com/CAMP-eXplain-AI/InputIBA

35th Conference on Neural Information Processing Systems (NeurIPS 2021).

bullet for feature attribution, each method is revealing a new aspect of model behavior and provides insights or new tools for getting closer to solving the problem of attribution.

Recently a promising solution – Information Bottleneck Attribution (IBA) [15] – grounded on information theory is proposed. The method explains the prediction via measuring the predictive information of latent features. This is achieved by placing an information bottleneck on the latent features. The predictive information of *input* features is then approximated via interpolation and averaging of latent features' information. The interpolated information values are considered as the importance of input features for the prediction. One shortcoming with this approach is the variational approximation inherent in the method, leads to an overestimation of information of features when applied to earlier layers. Another problem is that the interpolation to input dimension and the averaging across channels only approximates the predictive information of input features and is only valid in convolutional neural networks (CNNs) where the feature maps keep the spatial correspondences.

In this work, we propose InputIBA to measure the predictive information of *input* features. To this end, we first search for a bottleneck on the input that only passes input features that correspond to predictive deep features. The correspondence is established via a generative model, i.e. by finding a bottleneck variable that induces the same distribution of latent features as a bottleneck on the latent features. Subsequently, we use this bottleneck as a prior for finding an input bottleneck that keeps the mutual information with the output. Our methodology measures the information of input features with the same resolution as the input dimension. Therefore, the attributions are fine-grained. Moreover, our method does not assume any architecture-specific restrictions. The core idea – input bottleneck/mask estimation *using deep layers* – is the main contribution of this work to feature attribution research (input masking itself is already an established idea [16, 17], the novelty is leveraging *information of deep features* for finding the input mask).

We comprehensively evaluate InputIBA against other methods from different schools of thought. We compare with DeepSHAP [1] and Integrated Gradients [2, 3] from the school of Shapley value methods, Guided Backpropagation [5] (from backpropagation methods), Extremal perturbations [17] (from perturbation methods), GradCAM [18] (an attention-based method), and the background method, IBA [19]. We evaluate the method on visual (ImageNet classification) and natural language processing (IMDB sentiment analysis) tasks and from different perspectives. Sensitivity to parameter randomization is evaluated by Sanity Checks [12]. Evaluating the importance of features is done using Sensitivity-N [9], Remove-and-Retrain (ROAR) [10], and Insertion-Deletion [11]. To quantify the degree of being fine-grained in visual tasks we propose a localization-based metric, Effective Heat Ratios (EHR).

## 2   Related Work

### 2.1   Explaining Predictions via Feature Attribution

We focus on explaining the models for single inputs and their local neighborhood, i.e. local explanation [20]. We categorize attribution methods within the local explanation paradigm. Some methods can belong to more than one category such as IBA which leverages perturbation and latent features.

**Backpropagation-based:** [21, 22] linearly approximate the network and propose the gradient as attribution. Deconvolution [23], Guided Backpropagation [5], LRP [7], Excitation Backprop [8], DeepLIFT[6] use modified backpropagation rules.

**Shapley value:** Considering the network's function as a score function, and input features as players, we can assign contributions to features by computing the Shapley value. Due to complexity, many approximations such as DeepSHAP [1], Integrated Gradients [2, 3] are proposed.

**Perturbation-based:** These methods perturb the input and observe its effect on the output value [16, 17, 24]. E.g. Extremal Perturbations[17] searches for the largest smooth mask on the input such that the remaining features keep the target prediction. LIME [25] linearly approximates the model around local input perturbations.

**Latent Features:** CAM/GradCAM [26, 18] use the activation values of final convolutional layers. IBA [19] measures the predictive information of latent features. Pathway Gradient [27] leverages critical pathways.

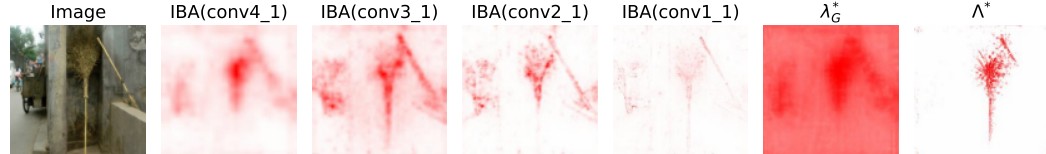

Figure 1: **Effect of** $P(Z)$ **Approximation (IBA [19] vs InputIBA ($\Lambda^*$)):** We see the result of applying IBA on different layers of a VGG16 network (from conv4_1 to conv1_1). The approximations (averaging across channels) in IBA result in the assignment of information to irrelevant areas of the image (the trashcan and areas around the image). As we move towards earlier layers (conv1_1), the information is distributed equally between features, and less information is assigned to the most relevant feature (the broom), due to the overestimation of mutual information $I[R, Z]$ in IBA. Using our approximation of $P(Z)$ the resulting mask $\Lambda^*$ is representing only the relevant information (the broom) at input resolution. $\lambda_G$ represents the prior knowledge we use for $P(Z)$. IBA uses the Gaussian distribution ($Q(Z) \sim \mathcal{N}(\mu_R, \sigma_R)$) to approximate $P(Z)$.

## 2.2   Do Explanations Really Explain?

The story starts with Nie et al. [13] demonstrating that Deconvolution [23] and Guided Backpropagation [5] are reconstructing image features rather than explaining the prediction. Later Adebayo et al. [12] propose sanity check experiments to provide further evidence. They show some methods generate the same attribution when the weights of the model are randomized.

Sixt et al. [14] and Khakzar et al. [28, 29] provide further proofs and experiments, and add other attribution methods to the circle. It is noteworthy that all these methods generate visually interpretable results. In parallel, several feature importance evaluations are proposed to evaluate whether the features recognized as important, are indeed contributing to the prediction. All these experiments are grounded on the intuition that, removing (perturbing) an important feature should affect the output function relatively more than other features. These methods are Remove-and-Retrain [10], Sensitivity-N [9], Insertion/Deletion [11], which are introduced in Section 4. Interestingly, such evaluations reveal that axiomatic and theoretically iron-clad methods such as Shapley value approximation methods (e.g. Integrated Gradients [2, 3] and DeepSHAP [1]) score relatively low in these feature evaluation methods.

## 3   Methodology

We first introduce the background method, IBA [15] in Section 3.1. We proceed in Section 3.2 with explaining IBA's shortcomings and proposing our solution. In Section 3.3 and Section 3.4 we explain details of our solution.

### 3.1   Background - Information Bottleneck Attribution (IBA)  [15]

This method places a bottleneck $Z$ on features $R$ of an intermediate layer by injecting noise to $R$ in order to restrict the flow of information. The bottleneck variable is $Z = \lambda R + (1 - \lambda)\epsilon$, where $\epsilon$ denotes the noise and $\lambda$ controls the injection of the noise. $\lambda$ has the same dimension as $R$ and its elements are in $[0, 1]$. Given a specific input $I$ and its corresponding feature map $R$ ($R = f(I)$, function $f$ represents the neural network up to the hidden layer of R), the method aims to remove as many features in $R$ as possible by adding noise to them (via optimizing on $\lambda$), while keeping the target output. Therefore, only features with predictive information will pass through the bottleneck (parameterized by the mask $\lambda$). The bottleneck is thus optimized such that the mutual information between the features $R$ and noise-injected features $Z$ is reduced while the mutual information between the noise-injected features $Z$ and the target output $Y$ is maximized, i.e.

$$\max_{\lambda} I[Y, Z] - \beta I[R, Z] \tag{1}$$

where,

$$I[R, Z] = E_R[D_{KL}[P(Z|R)||P(Z)]] \tag{2}$$

## 3.2 Approximating the Distribution of Bottleneck P(Z)

The main challenge with computing $I[R, Z]$ is that the distribution $P(Z)$ is not tractable as we need to integrate over all possible values of $R$ (since $P(Z) = \int P(Z|R)P(R)dR$). Therefore IBA methodology resorts to variational approximation $Q(Z) \sim \mathcal{N}(\mu_R, \sigma_R)$. The assumption is reasonable for deeper layers of the network [19]. However, as we move toward the input, the assumption leads to an over-estimation of mutual information $I[R, Z]$ [19]. The result of using such an approximation $(Q(Z) \sim \mathcal{N}(\mu_R, \sigma_R))$ is presented in Fig. 1 for various layers of a VGG-16 neural network. As the over-estimation of $I[R, Z]$ increases by moving towards the earlier layers, the optimization in Eq. (1) removes more features with noise. We can see that the predictive feature (the broom) in Fig. 1 disappears as we move IBA towards the early layers (to `conv1_1`). The approximation of $P(Z)$ is most accurate when IBA is applied on the deepest hidden layer.

In order to accomplish feature attribution for the *input* features, IBA method limits itself to convolutional architectures. First, it finds the informative features of a hidden layer by solving Eq. (1). Let $\lambda^*$ denote the result of this optimization. IBA interpolates these values to the input dimension (similar to CAM [26]) and averages $\lambda^*$ across channel dimension. Such interpolation is reasonable only for convolutional architectures as they keep spatial information. The interpolation and averaging introduce a further approximation into computing predictive information of input features. From this perspective, it is desirable to apply IBA to early layers to mitigate the effect of interpolation. At the same time, early layers impose an overestimation of $I[R, Z]$. Therefore, there is a trade-off between mitigating the adverse effect of interpolation/averaging and the $Q(Z) \sim \mathcal{N}(\mu_R, \sigma_R)$ approximation.

We aim to come up with a more reasonable choice for $Q(Z)$ such that it is applicable for the input $I$. This alleviates architecture dependency to CNNs, as the mutual information is directly computed on the input space and avoids the approximation error which results from interpolation to input dimension and the summation across channels. We take advantage of $Q(Z) \sim \mathcal{N}(\mu_R, \sigma_R)$ being reasonable for deep layers. Thus we first find a bottleneck variable $Z^*$ parameterized by the optimal result $\lambda^*$ ($Z^* = \lambda^* R + (1 - \lambda^*)\epsilon$ ) by solving Eq. (1). The bottleneck $Z^*$ restricts the flow of information through the network and keeps deep features with predictive information. Then, we search for a bottleneck variable on input, $Z_G$, that corresponds to $Z^*$ in the sense that applying $Z_G$ on input inflicts $Z^*$ on the deep layer. This translates to finding a mask on input features that admits input features which correspond to informative deep features. Thus, the goal is to find $Z_G$ such that $P(f(Z_G)) = P(Z^*)$, where function $f$ is the neural network function up to feature map $R$:

$$\min_{\lambda_G} D[P(f(Z_G))||P(Z^*)] \tag{3}$$

where $Z_G = \lambda_G I + (1 - \lambda_G)\epsilon_G$, and $D$ represents the distance similarity metric. For details of Eq. (3) refer to Section 3.3.

The resulting $Z_G^*$ (and its corresponding $\lambda_G^*$) from Eq. (3) is a bottleneck variable on input $I$ that keeps input features associated with predictive deep features. The final goal is keeping only predictive features of input (Eq. (1)), therefore, $Z_G$ can be used as prior knowledge for the distribution of input bottleneck $P(Z_I)$. To incorporate this prior, we condition $P(Z_I)$ on $Z_G$, i.e. $Z_I = \Lambda Z_G + (1 - \Lambda)\epsilon$ with a learnable parameter $\Lambda$ as the input mask. We then proceed and solve Eq. (1) again, but this time for $Z_I$. The resulting mask from this optimization is denoted by $\Lambda^*$. We refer to this methodology resulting in $\Lambda^*$ as **InputIBA**.

**Remark 1** $P(Z_I) \sim \mathcal{N}(\lambda_G \Lambda I + (1 - \lambda_G \Lambda)\mu_I, (1 - \lambda_G \Lambda)^2 \sigma_I^2)$

**Remark 2** $P(Z_I|I) \sim \mathcal{N}(\Lambda I + (1 - \Lambda)\mu_I, (1 - \Lambda)^2 \sigma_I^2)$

We can explicitly compute $D_{KL}[P(Z_I|I)||P(Z_I)]$ in order to compute mutual information $I[Z_I, I]$, the assumptions and detailed derivation are provided in the Appendix A.

**Proposition 3** *For each element $k$ of input $I$ we have* $D_{KL}[P(Z_{I,k}|I_k)||P(Z_{I,k})] = \log \frac{1-\lambda_{G,k}\Lambda_k}{1-\Lambda_k} + \frac{(1-\Lambda_k)^2}{2(1-\lambda_{G,k}\Lambda_k)^2} + \frac{(I_k - \mu_{I,k})^2(\Lambda_k - \lambda_{G,k}\Lambda_k)^2}{2(1-\lambda_{G,k}\Lambda_k)^2\sigma_{I,k}^2} - \frac{1}{2}$

## 3.3 Estimating the Input Bottleneck $Z_G$

The objective is to find a bottleneck variable $Z_G$ at the input level that inflicts the distribution $P(Z^*)$ on the latent layer where $Z^*$ is computed. In other words, we search for $\lambda_G$ that minimizes

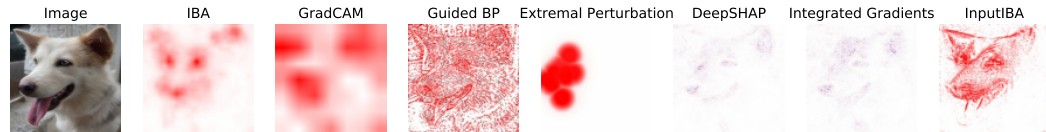

| Image | IBA | GradCAM | Guided BP | Extremal Perturbation | DeepSHAP | Integrated Gradients | InputIBA |

Figure 2: **Qualitative Comparison (ImageNet)**: We observe that the results of different attribution methods are substantially dissimilar for the same prediction. This is a caveat for the community that the attribution problem is far from solved. There is a consensus between GradCAM, IBA, and our method (InputIBA) that the ears and the snout are important for the prediction. Feature importance evaluations show that these methods reveal important features. InputIBA, DeepSHAP and Integrated Gradients are fine-grained. However, we observe in feature importance experiments that only the InputIBA is attributing to important features among fine-grained methods.

$D[P(f(Z_G))||P(Z^*)]$, where function $f$ is the neural network function before bottleneck $Z^*$, and $D$ is the similarity metric. We employ a generative adversarial optimization scheme to minimize the distance. Specifically, the generative adversarial model we use tries to fit the two distribution with respect to Wasserstein Distance ($D$) [30]. In this work we focus on local explanation, i.e. given a sample input, we aim to find an explanation that is valid for the neighbouring area of sample input ($B_\epsilon(R) := \{x \in R^n : d(x, R) < \epsilon\}$). We refer to this set, as the local explanation set. To generate $f(Z_G)$, we sample $I$ from the local explanation set of the input. Variable $Z_G = \lambda_G I + (1 - \lambda_G)\epsilon_G$ is constructed from $\lambda_G, \mu_G$ and, $\sigma_G$. These are the learnable parameters of our generative model. We apply the reparametrization trick during sampling learnable noise $\epsilon_G$ (parameterized by mean $\mu_G$ and standard deviation $\sigma_G$), so that the gradient can be backpropagated to random variable $\epsilon_G$. Once we have the sampled values ($I, \mu_G$ and $\sigma_G$) we get $Z_G = \lambda_G I + (1 - \lambda_G)\epsilon_G$, we pass this variable through the neural network to get a sample of $f(Z_G)$. The target distribution that the generator is approximating is the $P(Z^*)$. We construct the target dataset by sampling from $P(Z^*)$. The generative adversarial model also leverages a discriminator which is learned concurrently with the generator. The discriminator serves to discriminate samples from $P(f(Z_G))$ and $P(Z^*)$, thus its architecture can be determined based on the actual data type of the attribution sample.

### 3.4   Optimizing I[Y,Z]

Minimizing cross-entropy loss ($\mathcal{L}_{CE}$) is equivalent to maximizing the lower bound of the mutual information $I[Y, Z]$ [31]. Thus, in this work and in [19], instead of optimizing Eq. (1) we optimize:

$$\min_\lambda \mathcal{L}_{CE} + \beta I[R, Z] \tag{4}$$

In this paper, we provide more support for using cross-entropy instead of $I[Y, Z]$. Given the network with parameter set $\theta$ denoted as $\Phi_\theta$, we derive the exact representation of $I[Y, Z]$ instead of the upper bound of $I[Y, Z]$, Full derivation can be found in Appendix B.

**Proposition 4** *Denoting the neural network function with $\Phi_\theta$, we have:*
$I[Y, Z] = \int p(Y, Z) \log \frac{\Phi_\theta(Y|Z)}{p(Y)} dY dZ + \mathbb{E}_{Z \sim p(Z)}[D_{KL}[p(Y|Z)||\Phi_\theta(Y|Z)]]$

We prove that the minimizer of the cross-entropy loss is the maximizer of the mutual information $I[Y, Z]$ in the local explanation setting (exact assumption and proof provided in the Appendix C).

**Theorem 5** *For local explanation (local neighborhood around the input),*
$\arg\max \int p(Y, Z) \log \frac{\Phi_\theta(Y|Z)}{p(Y)} dY dZ = \arg\max I[Y, Z]$

## 4   Experiments and Results

Over the course of this section, first we provide the experimental setup in Section 4.1. Then, we present qualitative results in Section 4.2, and check the InputIBA's sensitivity to parameter randomization in Section 4.3. We proceed with evaluation of the attribution methods in terms of human-agnostic feature importance metrics (Sensitivity-N [9] in Section 4.4.1, Insertion/Deletion [11], and ROAR [10]). Finally, we evaluate the methods in terms of localization Section 4.5 using

| Method | Text (Tokens Divided by Space) |
| --- | --- |
| InputIBA | this is easily and clearly the best . it features loads of cameos by big named comedic stars of the age , a solid script , and some great disneyesque songs , and blends them together in a culmination of the best display of henson ' s talent . |
| IBA | this is easily and clearly the best . it features loads of cameos by big named comedic stars of the age , a solid script , and some great disneyesque songs , and blends them together in a culmination of the best display of henson ' s talent . |
| LIME | this is easily and clearly the best . it features loads of cameos by big named comedic stars of the age , a solid script , and some great disneyesque songs , and blends them together in a culmination of the best display of henson ' s talent . |
| Integrated Gradients | this is easily and clearly the best . it features loads of cameos by big named comedic stars of the age , a solid script , and some great disneyesque songs , and blends them together in a culmination of the best display of henson ' s talent . |

Table 1: **Qualitative Comparison (IMDB)**: We compare with IBA and methods that are widely used for NLP model interpretaion. To visualize the attribution, we highlight words that have attribution values from 0.33 to 0.66 with orange, and words that have attribution values from 0.66 to 1 with red (attribution value ranges from 0 to 1). We see that IBA does not translate to RNN architectures when attributing to input tokens. IBA can only identify latent important features. We observe that InputIBA and LIME, and IG attribute to relevant tokens.

our proposed EHR metric. To demonstrate the model-agnostic capability and show that our model is revealing the predictive information, we evaluate our method in both vision and NLP domains. We choose ImageNet [32] image classification for evaluation on vision domain. Apart from image classification tasks on ImageNet/VGG-16, we also apply InputIBA on a sentiment classification task (IMDB) and an RNN architecture.

## 4.1 Experimental Setup

In our experiments on ImageNet, we insert the information bottleneck at layer `conv4_1` of VGG16 [33] pre-trained by `Torchvision` [34]. We use $\beta_{\text{feat}} = 10$ for IBA. We adopt Adam [35] as the optimizer with learning rate of $1.0$, and optimize the feature bottleneck for 10 optimization steps. We optimize the generative adversary model for 20 epochs with RMSProp [36] optimizer, setting the learning rate to $5 \times 10^{-5}$. For optimizing $\Lambda$, we use $\beta_{\text{input}} = 20$ and run 60 iterations. The choice of hyper-parmeters of attribution methods affects their correctness [37]. For more details on the hyper-parameters please refer to Appendix D.

In the NLP experiment, we investigate the sentiment analysis task. A 4-layer LSTM model is trained on IMDB dataset [38] which consists of 50000 movie reviews labeled as either "positive" or "negative". After the model is trained, we generate attributions at embedding space. We compare InputIBA with 4 baselines: Integrated Gradients, LIME [20], IBA [15] and random attribution. In order to generate attribution using IBA, we insert IBA at a hidden layer and consider the hidden attribution as valid for the embedding space, since RNN does not change the size and dimension of input tensor.

We train our model on a single NVIDIA RTX3090 24GB GPU. For VGG16 [33], it takes 24 seconds to obtain the attribution map of one image with resolution $224 \times 224$.

## 4.2 Qualitative Comparison

**ImageNet**  Fig. 2 presents the attribution resulting from various methods. IBA and GradCAM are interpolated into input space, thus they are blurry. Guided Backpropagation is reconstructing image features [13], thus looks interpretable. Extremal Perturbation has lost the shape information of the target object as it applies a smooth mask, moreover it has converged to a region (dog's snout) that is sufficient for the prediction. DeepSHAP and Integrated Gradients generate fine-grained attributions, however, the negative/positive (blue/red) assignment seems random. Moreover, feature

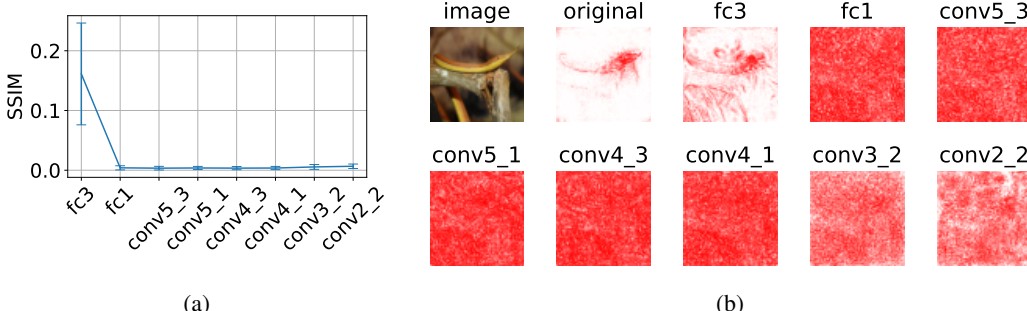

(a)                                                          (b)

Figure 3: **Parameter Randomization Sanity Check [12]:** The experiments evaluate whether the attribution changes after randomizing the parameters of neural networks. The randomization starts from FC3 layer and moves towards the image (a) Correlation between original attribution and the attribution after randomization (results averaged for ImageNet subset). We observe that the correlation rapidly moves to 0. The error bars denote $\pm 1$ standard deviation; (b) Attribution map before randomization (label "original") and after randomization the parameters up the specified layer. We observe that attribution is also randomized after parameters are randomized (as opposed to methods such as Guided Backpropagation and LRP (Deep Taylor variant) [12, 14]).

importance experiments (Section 4.4) show the highlighted features are not important for the model. Our proposed method is fine-grained and visually interpretable while highlighting important features (Section 4.4). More qualitative examples (randomly selected) are provided in Appendix E.

**IMDB**    Table 1 presents attribution on a text sample from IMDB. We see that IBA fails to generate a reasonable explanation by assigning high attribution to all words. The observation also shows that IBA is not a model-agnostic method, since IBA assumes the spatial dependency between input and hidden space, which doesn't hold for recurrent neural networks.

### 4.3   Parameter Randomization Sanity Check [12]

The purpose of this experiment is to check whether the attribution changes if model parameters are randomized. If the attribution remains unchanged, then the attribution method is not explaining model behavior. The experiment progressively randomizes the layers starting from the last layer, and generating an attribution at each randomization step. The generated attribution is compared in terms of similarity with the original attribution.

The similarity is measured in terms of structural similarity index metric (SSIM) [39]. In this experiment, we randomly select 1000 samples, and compute the average of the SSIM for all the samples. Fig. 3a, Fig. 3b demonstrate the SSIM error bars and an example for visual inspection respectively. We observe that the InputIBA is sensitive to this randomization (Fig. 3b).

### 4.4   Feature Importance Evaluation

The following experiments evaluate whether the features identified as important by each attribution method are indeed important for the model.

#### 4.4.1   Sensitivity-N [9]

The experiment randomly masks $n$ pixels and observes the change in the prediction. Subsequently, it measures the correlation between this output change and the sum of attribution values inside the mask. The correlation is computed using Pearson Correlation Coefficient (PCC). The experiment is repeated for various values of $n$. For each value $n$ we average the result from several samples. For ImageNet, we run this experiment on 1000 images. We randomly sample 200 index sets for each $n$. In the IMDB dataset, texts have different lengths, thus we cannot perturb a fixed number of words across samples. We address this issue by slightly changing the method to perturb a fixed percentage of words in a text. On ImageNet (Fig. 4) the InputIBA outperforms all baseline methods when we perturb more than $10^2$ pixels. When evaluating on IMDB dataset, the InputIBA exhibits a higher

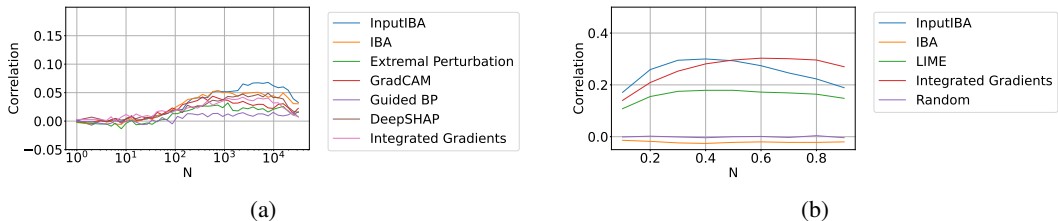

(a)                                          (b)

Figure 4: **Feature Importance - Sensitivity-N**: (a) ImageNet dataset: InputIBA, IBA and GradCAM show high scores in this feature importance metric. We observe that fine-grained (and also axiomatic) methods such as DeepSHAP and Integrated Gradients achieve relatively low scores. (b) IMDB dataset: Both LIME and InputIBA identify important features according to this metric.

correlation under 50% text perturbation rate, which implies the InputIBA is performing better in not assigning attribution to irrelevant tokens.

### 4.4.2   Insertion/Deletion [11]

Deletion, successively deletes input elements (pixels/tokens) by replacing them with a baseline value (zero for pixels, <unk> for tokens) according to their attribution score. Insertion, inserts pixels/tokens gradually into a baseline input (image/text). In both experiments we start inserting or deleting the pixel/token with highest attribution. For each image, at each step of Insertion/Deletion we compute the output of the network, and then compute the area under the curve (AUC) value of the output for all steps on a single input. Then we average the AUCs for all inputs (On ImageNet, we average the AUC of 2000 images). For Insertion experiment, higher AUC means that important elements are inserted first. For Deletion experiment, lower AUC means important elements were deleted first. The results are presented in Fig. 5. For the vision task (ImageNet) InputIBA outperforms the rest. Note that the axiomatic methods DeepSHAP and Integrated Gradients achieve low scores in both Deletion/Insertion on ImageNet.

### 4.4.3   Remove-and-Retrain (ROAR) [10]

One underlying issue with Sensitivity-N and Insertion/Deletion experiments is that the output change may be the result of model not having seen the data during the training. Therefore, ROAR [10] retrains the model on the perturbed dataset. The more the accuracy drops the more important the perturbed features are. For each attribution method, the perturbation of the dataset is done from the most important elements (according to attribution values) to the least important element. As retraining the model is required for each attribution method at each perturbation step, the experiment is computationally expensive. Therefore, we run this experiment on CIFAR10. The results are presented in Fig. 6. We can observe two groups of methods. For the first group (DeepSHAP, Integrated Gradients and Guided Backpropagation) the accuracy does not drop until $70\%$ perturbation, meaning if $70\%$ of pixels are removed, the model can still have the original performance. For the rest of the methods, we see that the features are indeed important. GradCAM and Extremal perturbations are performing slightly better than IBA and InputIBA. We suspect that this is due to their smooth attribution maps. We test this hypothesis by applying smoothing on InputIBA (InputIBA*) and we observe that the performance becomes similar to the other two. The key observation is that these four methods are all successful in identifying important features.

### 4.5   Quantitative Visual Evaluation via Effective Heat Ratios (EHR)

We quantify the visual alignment between attribution maps and ground truth bounding boxes (on ImageNet [32]). This serves as a metric for *visual/human* interpretability, and is a *measure of fine-grainedness*. Previously [19], the proportion of top-$n$ scored pixels located within the bounding box was computed. We argue that this method only considers the ranks of the pixels, but ignores the distribution of attribution outside the bounding box. We illustrate the limitation of the previous metric with synthetic examples in Appendix F. Instead, we suggest to vary the value of $n$ to consider the distribution of the attribution. To this end, we propose EHR, where we consider multiple quantile thresholds from 0.0 to 1.0. At each quantile threshold, we compute the sum of attribution within

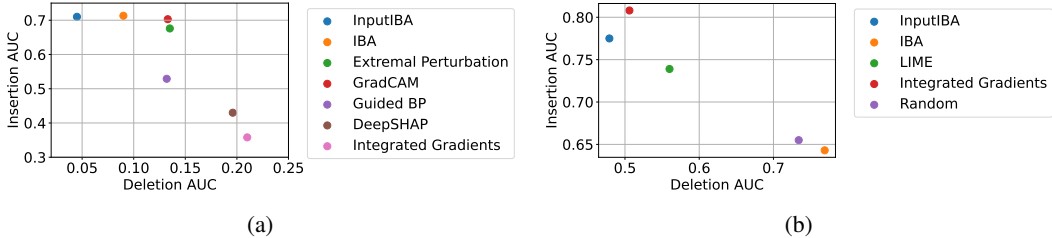

(a)                 (b)

Figure 5: **Feature Importance - Insertion/Deletion**: The closer the method is to the top-left the better (a) **ImageNet** dataset: InputIBA, IBA, Extremal Perturbations and GradCAM all reveal important features according to this metric. DeepSHAP and Integrated Gradients to do not idenfity important features. (b) **IMDB** dataset: InputIBA reveals important features according this metric. The results are consistent with Sensitivity-N.

| Method | EHR |
|---|---|
| InputIBA | $\mathbf{0.476} \pm 0.007$ |
| IBA | $0.356 \pm 0.005$ |
| GradCAM | $0.283 \pm 0.005$ |
| Guided BP | $0.421 \pm 0.005$ |
| Extremal Perturbation | $0.421 \pm 0.007$ |
| DeepSHAP | $0.183 \pm 0.002$ |
| Integrated Gradients | $0.155 \pm 0.002$ |

Table 2: **Quantitative Visual Evaluation - EHR**: This metric evaluates how precisely the attributions localize the features by comparing them with ground truth bounding boxes. InputIBA, extremal perturbations, and Guided Backpropagation score highest. IBA and Grad-CAM also perform well, but due to their lower resolution maps, they receive lower scores. Standard error is also presented in the table.

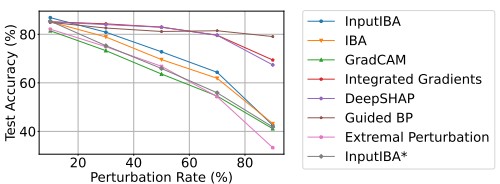

Figure 6: **Feature Importance - ROAR**: Integrated Gradients, DeepSHAP, and Guided Back-propagation are not identifying contributing features. On the other hand, InputIBA, IBA, Extremal Perturbations, and GradCAM point to important features. The latter two methods are performing slightly better, which is due to their attributions being smooth (covering more areas). We apply smoothing to InputIBA (InputIBA*) and achieve the same result.

the bounding box divided by the total number of pixels above this threshold, which is defined as the *effective heat ratio*. Finally, we compute the AUC of the ratios over the quantiles. We assess InputIBA along with other baselines on 1000 images with bounding box covering less than 33% of the whole image. Table 2 illustrates the results.

### 4.6 Discussion

**Societal Impact**    Although our work is a step towards solving the attribution problem, the problem remains open. Therefore the interpretation tools (attribution methods) must be used with caution. The machine learning community is using these tools to interpret their results and is deriving conclusions. It is not clear if the findings would also show up with another interpretation tool. Wrong interpretations of results arising from the attribution tools can have a destructive effect in mission-critical applications such as medical imaging.

The community must be aware of the shortcomings of each interpretation tool. For example, [13, 14] prove several methods are not explaining the behavior. In our work, we also observe that two Shapley value-based methods with solid mathematical and axiomatic grounding, are not identifying important features on computer vision datasets according to all three feature importance evaluations (this also verified for Integrated Gradients in [10]). The Shapley value is hailed as the ultimate solution by many research works, our observation is telling another story. Therefore, our proposed method (and IBA [19]) should also be used with caution, even though they are grounded on theory and are performing well in current metrics.

**Limitations**    Although our core idea – finding a bottleneck on input that corresponds to predictive deep features (described in Section 3.3 ) – is simple, the entire methodology for computing the

predictive information of input features is relatively complex (compared to a method such as CAM [26, 18]). This may be an impediment to the method's adoption by the community (however, we provide a simple interface in our code).

Another issue is that our idea adds an additional optimization term (Eq. (3)) to IBA, which increases the time complexity. Therefore, our method is introducing a trade-off between improved performance and speed. Nonetheless, for many interpretation applications, speed may not be an issue.

# 5   Conclusion

In this work, we propose a methodology to identify the *input* features with high predictive information for the neural network. The method is agnostic to network architecture and enables fine-grained attribution as the mutual information is directly computed in the input domain. The improved interpretation technique benefits the general machine learning community. Furthermore, it introduces a new idea – input bottleneck/mask estimation *using deep layers* – to the attribution research community to build upon and move one step closer to solving the attribution problem.

**Acknowledgement**   Ashkan Khakzar and Azade Farshad are supported by Munich Center for Machine Learning (MCML) with funding from the Bundesministerium fur Bildung und Forschung (BMBF) under the project 01IS18036B. Yawei Li is supported by the German Federal Ministry of Health (2520DAT920). Seong Tae Kim is supported by the IITP Grant funded by the Korea Government (MSIT) (Artificial Intelligence Innovation Hub) under Grant 2021-0-02068.

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
