# OpenReview forum: "Fine-Grained Neural Network Explanation by Identifying Input Features with Predictive Information"
_NeurIPS.cc/2021/Conference — NeurIPS 2021 Poster_

### Official Review · Reviewer_auMi · 2021-07-16

**Rating:** 6
**Confidence:** 3

**Summary:**

Authors build up on a recent attribution technique (IBA), identifies shortcomings and proposes a new technique that utilizes a measure of information for input features (vs latent). New method does not have architecture restrictions.

They evaluate their method with other types of explanability techniques in a set of image and NLP tasks and propose an additional task/metric with "effective heat ratio" (EHR).


**Limitations And Societal Impact:**

Adequately addressed.

**Main Review:**

Work is novel, clear, and further the field of XAI. The work is somewhat original (building heavily on IBA). Experiments are extensive enough for image and text data. I am also glad the author have done sanity checks on their technique. I rate the work a 6 ("Marginally above the acceptance threshold") and think with a few edits it can be a 7 or 8.

Authors refer to their method as "Ours" or as "Λ", authors might want to consider a name for their method so that future work can more easily reference it (InputBA?).

The authors mention in line 310 that mask estimation is a new idea to the attribution community. This might be overclaimed since GNN explainability techniques will often figure out a mask for the inputs [1].

The hyperparameters of an explainability technique have a strong influence on their performance [2], authors might want to expand on the hyperparameters of "their" technique.

Work could be improved by including other data modalities (tabular data, or graph data [4]) and evaluating them quantitatively [4-8]. In particular, the lens of explanation qualities might be useful to talk about failure modes or strengths of different explainability techniques ([3]).

Confidence Intervals or statistical tests on numerical results would strengthen claims.


Citations:
[1] https://arxiv.org/abs/1903.03894
[2] https://arxiv.org/abs/2003.08754
[3] https://link.springer.com/chapter/10.1007/978-3-319-90403-0_9
[4] https://papers.nips.cc/paper/2020/hash/417fbbf2e9d5a28a855a11894b2e795a-Abstract.html
[5] https://arxiv.org/abs/1904.11829
[6] https://arxiv.org/abs/2003.07258
[7] https://arxiv.org/abs/1806.10758
[8] https://arxiv.org/abs/1806.10758

**Time Spent Reviewing:**

2 hrs

---

> ### Author Response · Authors · 2021-08-09
> **Addressing the suggestions and comments of reviewer auMi**
>
> We sincerely thank the reviewer for the suggestions to improve the work and for acknowledging the novelty. In the following, we try to address all the raised points.
>
> (**We use \* for the references provided by the reviewer to avoid confusion.** E.g. [2\*] denotes reference number 2 provided by the reviewer.)
>
> ---
>
> ##### **1) “Authors refer to their method as "Ours" or as "Λ", authors might want to consider a name for their method so that future work can more easily reference it (InputBA?).”**
>
> We appreciate the remark. We intended to give tribute to the original IBA method and emphasize that our propositions are for improving explanations based on IBA. Your proposition (InputIBA or InputBA) also serves this purpose and we can use this name.
>
> ---
>
> ##### **2) “The authors mention in line 310 that mask estimation is a new idea to the attribution community. This might be overclaimed since GNN explainability techniques will often figure out a mask for the inputs.”**
>
> In the conclusion [line 310] we claim mask estimation using **deep layers** is a new idea. Applying input masking is indeed common and one of the main approaches within the perturbation-based approach of explanation as stated in related work (Sec. 2.1, lines 66-67). The provided reference [1\*] (GNNexplainer) is also a masking method, and before that, the following methods also applied masking strategy: Meaningful perturbation [36], extremal perturbations [17], ...
>
> However, in all of these works, the methods directly search for a mask on the input and use the output during the optimization. Performing the optimization without restriction can end up in trivial results [17,36] as the space of solutions is huge. For a simple case, finding the smallest mask that keeps the output the same, the optimization can simply result in a trivial adversarial case [36]. Some of these [17, 36] methods add certain constraints during optimization (e.g. finding only smooth masks) to avoid this issue. Our proposed methodology is guided by the informative (important) deep features, **thus the input mask is limited to the space of masks that only keep these high-level important features. To our knowledge, this is a novel approach that we contribute to the input masking methodologies**. As the space of input masks is restricted to the ones that keep important deep features, we remove the possibility of finding many of the trivial masks during optimization.
>
> Our proposed deep layer restriction can indeed be extended to GNNExplainer, Extremal perturbations, and others. We appreciate the comment, we will definitely add this discussion to make this contribution clear.
>
> [36] Interpretable Explanations of Black Boxes by Meaningful Perturbation
>
> ---
>
> ##### **3) “The hyperparameters of an explainability technique have a strong influence on their performance [2\*], authors might want to expand on the hyperparameters of "their" technique.”**
>
> We are very grateful for this remark and the provided reference. Indeed further discussion of this aspect of the work is very insightful.
> The hyperparameters and model architectures we used for our method are provided in Sec. 4.1 and Appendix D.1. We will add qualitative examples to display the effect of our hyperparameters in the Appendix (due to page limit). Our exclusive hyperparameters are the number of data samples from $Z^{\*}$ used for estimating $Z_{G}$ and the beta value when we solve Eq. 1 for $Z_{I}$ for which can provide further details. We will use the provided reference [2\*] work to more strongly emphasize the importance of hyperparameters for the readers in the main manuscript.
>
> ---
>
> ##### **4) “Work could be improved by including other data modalities (tabular data, or graph data [4\*]) and evaluating them quantitatively [4\*-8\*]. In particular, the lens of explanation qualities might be useful to talk about failure modes or strengths of different explainability techniques ([3\*]).”**
>
> Very useful suggestion. As also noted by the reviewer, the method is not restricted to specific architectures. We are currently planning to extend the work to the graph domain. Due to the many nuances existing in the problem, it will be an independent work. For instance, in graphs, we have different cases of transductive and inductive graphs. Moreover, in graphs there is the possibility to find the importance of edges in addition to nodes. We are also considering both node-level and graph-level classification cases. Therefore, we did not discuss graph data in this work. We would like to mention that we are considering NLP alongside vision (though the experiments emphasize the vision task), whereas it is common to evaluate only one task. We made an effort to add many experiments to provide new findings for all methods for future reference.
>
> The specific advantage that our method will bring to graph explanation, is the contribution that we use the information in deep layer features to find the input mask. As explained in point 2, searching for a mask on the input without restricting the space of solutions can result in trivial (e.g. adversarial) masks. Moreover, due to the architectural independence, our method would not be restricted to certain structures such as the CAM-inspired method [38], where the method is only applicable to graph level classification and can only highlight nodes. These details require a separate manuscript, which is probably the opinion of the reviewer as well as we understand.
>
> Regarding suggested quantitative experiments (references [4\*-8\*]): We also use the [7\*,8\*] experiments in Sec. 4.4.3, the paper is the work that introduces ROAR. [4\*] is proposed for Graphs as the reviewer states and [5\*] for RNNs. Their ideas indeed can be useful in a future work. The [6\*] could suffer from the following issue. The method uses a generated dataset, and trains a model on it, and expects the model to use these features. It is possible that the model uses other features (not expected by us). Thus a correct attribution method may be introduced as problematic within this framework. Nevertheless, we have made an effort to extensively evaluate our method using five established quantitative metrics: Sanity Checks, Sensitivity-N, Insertion/Deletion, EHR, and ROAR [7\*,8\*].
>
> [38] Explainability Methods for Graph Convolutional Neural Networks
>
> ---
>
> ##### **5) “Confidence Intervals or statistical tests on numerical results would strengthen claims.”**
>
> We agree that it would strengthen the claims, though it has not been a requirement in the literature of explanation methods. Nevertheless, we have run some experiments such as ROAR and Insertion/Deletion multiple times and reported the mean value (in order to have clean and easy-to-read plots). For Sanity Checks, we have included std in the plots (Fig. 3). We can definitely add details such as mean, std, and statistical tests.

---

### Official Review · Reviewer_DbuT · 2021-07-16

**Rating:** 8
**Confidence:** 3

**Summary:**

This paper introduces a new interpretability method, building on top of the Information Bottleneck Attribution [IBA] literature, by ensuring that the predictive information of the input features is extracted directly, rather than through approximation as in IBA. This method has the advantage that no variational approximation is needed for the input probability distribution, meaning that the interpolation to the input dimension is also not needed. The contribution of this method is that it provides accurate info about the interpretation offered by the input features due to lack of overapproximation. The paper also contains detailed experiments comparing multiple interpretation methods in the literature, using varied metrics and also passes the sanity check for sensitivity to parameter randomization.

**Limitations And Societal Impact:**

The paper somewhat sufficiently addresses the societal impact. However, towards the end of the paper, it is important and helpful to have commentary on the advantages of different interpretation methods (including the one proposed in this paper), so that future work has decent categorization of interpretation methods useful for different tasks.

**Main Review:**

The paper is well structured, and gives thorough background and motivation for the problem. It also contains plenty of literature review, allowing readers to understand the progression of the story of interpretability relatively well. The contributions are solid, and the experimental results demonstrate superiority of this method over IBA, and similarity to past methods by other metrics. Questions:

1. Line 124, why is it reasonable to take the optimal result for Q(Z) as opposed to the average result? Can the paper list the disadvantages of taking the optimal result vs the average result?
2. Line 152 is the first time local explainability is mentioned as a goal. I would expect that to be mentioned earlier in the paper when different interpretation methods are being compared in the literature search, and especially comment on whether different methods are local vs global and put the current work and results in context of the usefulness due to the global vs local nature.
3. While it is commendable that the paper is comparing work on two tasks -- NLP and vision, it does not seem fair to compare interpretation methods specifically created in the Image domain (GradCAM, Integrated Gradients etc not LIME) when applied to NLP. There should be commentary provided on which methods rely on the spatial correlation depicted in images, to allow the readers to fairly compare the interpretation methods in Vision and NLP. There should also be commentary on the advantages and disadvantages different methods depict when crossing domains.
4. Section 4.2, it would be helpful to see multiple qualitative egs, and also a rough estimate of the number of egs where the paper’s method seemed to outperform. It will also be helpful to know how the examples were selected, to ensure that no cherry picking was involved.
5. In Section 4.4, it will be helpful to have information on the advantages/ disadvantages of the different metrics applied, and what it means to evaluate the number of examples considered for each.
6. I find it odd that the dataset was suddenly changed in 4.4.3, due to computational limitations. I would expect CIFAR evaluation for other metrics as well, if this was necessary.


**Time Spent Reviewing:**

5

---

> ### Author Response · Authors · 2021-08-09
> **Addressing the questions of reviewer DbuT**
>
> We sincerely thank the reviewer for the time spent reviewing the work and for writing about several positive aspects of this work alongside the raised questions.
>
> ---
>
> ##### **1) “Line 124, why is it reasonable to take the optimal result for Q(Z) as opposed to the average result? Can the paper list the disadvantages of taking the optimal result vs the average result?”**
>
> In the following we discuss all the approximations existing in the IBA framework and discuss the effect (disadvantage) of each approximation (e.g. averaging) vs our solution:
>
> - Variational approximation of $Q(Z)$: In IBA formulation [15], it is assumed that $Q(Z)$ follows a normal distribution $Q(Z) \sim \mathcal{N}(\mu_{R},\sigma_{R})$ . This approximation is more reasonable in deep layers [18] [Lines 104-111] and using it for early layers results in overapproximation of mutual information, the effect of it is explained in [Lines 104-111] and Fig. 1. The disadvantage of this approximation in early layers is that due to over-approximation more features are removed [Lines 104-111] (the effect is visible in Fig. 1 as we progress from conv4_1 to conv1_1).
>
>     - Within this variational approximation, there is another hidden approximation for $\mu$ and $\sigma$, which we assume was not the question of the reviewer, Nevertheless, we can discuss it: The $\mu$ and $\sigma$ are selected by sampling images from the dataset, computing their feature maps $R$ and computing their $\mu$ and $\sigma$. The effect of the sampling dataset is negligible in practice [18], though theoretically it is not discussed so far.
>
> - In the final step, after the mutual information $I(X,Z)$ is computed: $Z$ is in the latent space, i.e. it is the masked version of an intermediate feature map $R$. The goal is computing the information of input features. But since $Z$ is not in input space, $I(X,Z)$ does not reflect the information of input features. IBA takes advantage of CNN’s property, as CNNs keep spatial correspondence between input and latent features (a certain activation is associated with its receptive field) and approximate the input information as such:
>
>     1. IBA averages the computed $I(X,Z)$ for the selected deep layer (which is the same dimension as $Z$) across the channel dimension and ends up with a 2-dimensional map: One disadvantage of this approximation is that, if a channel has high information, but the rest have close to zero information, the averaging reduces the value of the information at this spatial location. This results in the information map being smoothed out and areas where many channels have low information to have a relatively equal average to the previous case. Thus regions with highly informative features (the first case) get a similar score as regions with relatively little information.
>
>     2. The 2 dimensional $I(X,Z)$ is interpolated to input space: This results in a blurry information map, and many areas that do not have any information, get some values due to the interpolation from feature dimension e.g. (7*7) to input dimension
>
>     the effect (disadvantage) of these two approximations (1,2), averaging, and interpolation is visible in Fig. 1, where we compare IBA on a deep layer (conv4_1) to ours ($\Lambda^{*}$), where IBA is blurry and has information assigned to many areas in the input.
>
> ---
>
> ##### **2) “Line 152 is the first time local explainability is mentioned as a goal. I would expect that to be mentioned earlier in the paper …  comment on whether different methods are local vs global ...”**
>
> Thank you for the feedback. We agree that we should discuss local explainability earlier. Just to clarify, all the discussed works in the literature review and the compared methods are tackling the local explainability problem, we will make it clear in the manuscript.
>
> ---
>
> ##### **3) “While it is commendable that the paper is comparing work on two tasks -- NLP and vision, it does not seem fair to compare interpretation methods specifically created in the Image domain (GradCAM, Integrated Gradients etc not LIME) when applied to NLP … ”**
>
> We agree that a commentary on the advantages and disadvantages of methods for different tasks is insightful for the audience. For NLP task we did not use works that were introduced for the vision domain (e.g. extremal perturbations, GradCAM, GuidedBP). The exception is IBA, we used it on the NLP task to show our variation can be applied in NLP while the original does not. Integrated Gradients is also proposed for NLP in the original paper. We also used LIME as an instance of a perturbation-based method for the NLP domain. For the vision tasks we can claim that we have used an extensive set of methods compared to recent related works.
>
> + Integrated Gradients (IG): This method was originally proposed for both NLP and vision tasks. Later it was proven [3] that IG approximates the Shapley Value, which is the unique axiomatic solution to the credit assignment problem. We observe that this method works well on NLP but poorly in vision evaluations (as also reported in [10]).
> + DeepSHAP: It is another method that approximates Shapley value and it is shown that IG does a better approximation [3], therefore we only used the latter in NLP experiments. We used DeepSHAP in the vision task to show that it does not work well in this domain, since some of these evaluations (e.g. ROAR) were not reported earlier for DeepSHAP in the vision domain.
>
> ---
>
> ##### **4) “Section 4.2, it would be helpful to see multiple qualitative egs, and also a rough estimate of the number of egs where the paper’s method seemed to outperform. It will also be helpful to know how the examples were selected, to ensure that no cherry picking was involved.”**
>
> We provide more qualitative examples in Appendix E. Moreover, we have provided a jupyter notebook where researchers can surf through all examples.
> The intriguing observation (in Fig 1, Fig 2, and Appendix E) is that our method consistently has the advantages explained in the response to the first question. This is because our method uses the result of IBA to come up with a prior for distribution of $Z$ in the input space. Thus the results consistently become a more accurate version of IBA (without artifacts of averaging or interpolation). Results in the appendix are randomly selected, however, the main figures (Fig. 1 and 2) in the paper are selected to better reflect the effects of approximations in IBA. Nevertheless, qualitative examples are provided for depicting the advantages. In order to draw conclusions, we resort to quantitative experiments in Sec. 4.
>
> ---
>
> ##### **5) “In Section 4.4, it will be helpful to have information on the advantages/ disadvantages of the different metrics applied, and what it means to evaluate the number of examples considered for each.”**
>
> We have made an effort to explain the history and reasoning behind the experiments as they appeared in the literature. We grouped Sensitivity-N, Insertion/Deletion and ROAR together into feature importance metrics. In Sec. 4.4.3 (Line 255-257) we explain the advantage of ROAR over the other two. We value your suggestion and will further add explanations for the metrics differences. For instance, one problem with ROAR is the following: After the removal step there may remain other features that could be predictive for the model, and the model could pick them during retraining. Therefore, the accuracy would not drop, even though the explanation method might have accurately identified the discriminative features. While Sensitivity-N and Insertion/Deletion do not have this issue, they suffer from generating out of distribution samples due to perturbation. Due to the shortcomings of each experiment, it is imperative that the method be evaluated using all of them.
>
> ---
>
> ##### **6) “I find it odd that the dataset was suddenly changed in 4.4.3, due to computational limitations. I would expect CIFAR evaluation for other metrics as well, if this was necessary.”**
>
> Similar to literature, we have tried a challenging and standard dataset for the experiments, i.e. ImageNet [28]. In all these experiments, ImageNet provides a more reliable result, and the results of CIFAR can be redundant. However, running the experiment in Sec. 4.4.3, ROAR, requires enormous computational resources for ImageNet (multiple retraining steps for each attribution method on ImageNet, and for each input the attribution method is applied, and many attribution methods including ours require optimization for each input), thus recent works resort to CIFAR for this experiment or do not report any ROAR results. However, ROAR is very insightful as it shows whether the features identified by the method were indeed contributing to the output, and its results on CIFAR are still more insightful than not reporting it.

---

> > ### Comment · Reviewer_DbuT · 2021-08-30
> > **Response to Author Rebuttal**
> >
> > I truly appreciate the author's extensive response to my questions, and interest in further explaining their methods. I am happy with their responses, and have learned a lot during this process of reviewing. I'd like to change my review from "Good paper, accept" to "Top 50% of accepted NeurIPS papers, clear accept" after extensive discussion by the authors. I have reflected this change in my original review, though I have kept my questions there for reference of the discussion. Thank you for your efforts, authors!

---

### Official Review · Reviewer_MU86 · 2021-07-18

**Rating:** 7
**Confidence:** 4

**Summary:**

The paper presents a feature attribution method based on information bottleneck attribution (IBA). While IBA has shown to be an effective method for feature attribution, they are a few limitations such as, variational approximation. The authors mentioned that for lower layers of the network, the variational approximation causes an over-estimation of mutual information which leads to misleading attribution results. The authors proposed a bottleneck variable that keeps the deep features with prediction information and search for input that corresponds to the features in deep layers.

A thorough experimentation is conducted comparing vision and NLP models using various attribution methods belonging to a diverse set of classes. Quantitative results show that their method works better or on-par with the other attribution based methods.

**Limitations And Societal Impact:**

Yes, authors have done a great job in addressing the limitations of their approach.

**Main Review:**

The paper proposes an extension to information bottleneck attribution. Their proposed method is simple. However, it shows promising results when compared against several alternate methods.

Questions:
- Figure 2 shows full face of the object as important feature. I am wondering if the proposed method covers more area of the input? In other words, have a better recall in highlighting input features?
- Table 1: the results of IBA and integrated gradient are extremely bad. Is there a difference of scale of the attribution value which is causing IBA to highlight everything? For integrated gradient, what is the baseline? The confidence of the prediction score also have an effect on the input feature attribution. Did author look at them to make sure the they are running other methods with the optimal or reasonable settings?


**Time Spent Reviewing:**

3

---

> ### Author Response · Authors · 2021-08-09
> **Addressing the questions of reviewer MU86**
>
> We genuinely thank the reviewer for carefully going through the paper as reflected by the accurate summary, and thank the reviewer for appreciating the thorough experiments and promising results.
>
> We address the raised questions in the following:
>
> ---
>
> ##### **1) “Figure 2 shows full face of the object as important feature. I am wondering if the proposed method covers more area of the input? In other words, have a better recall in highlighting input features?”**
>
> Indeed an important question. One notion to bear in mind is that the model could be using any existing feature in the input for the prediction, and these features do not necessarily align with what humans use. Let’s take Fig. 2 itself as an example, depending on the training procedure (e.g. dataset, optimization objective,... ) the model may either use the ears, or the eyes, or the snout, the tongue, the fur texture, fur color, a combination of these features or might even use features that are imperceptible to us (there might be a pattern in intensity values, the dogs may be always facing left, ...). Therefore, if an attribution method covers more areas of human-interpretable features, or is aligned with what humans think are important features it does not necessarily mean it is correctly identifying what features the model is using. The purpose of the explanation method is to accurately reveal which feature the *model* uses.
>
> How do we know if the identified areas are the features that the model uses? One way is to run sanity checks [12] and observe which method is incorrect. This experiment shows that some methods such as Guided Backpropagation do not explain the model while having human-interpretable results (Sec 2.2). They generate the same saliency when the model is randomized. It is shown [13,14] that such methods reconstruct image features, and thus look interpretable to us. The literature also proposes feature importance experiments based on removing features and observing the output change of the model, such as Sensitivity-N (Sec. 4.4.1)[9], Insertion/Deletion (Sec. 4.4.2)[11], and ROAR (Sec. 4.4.3)[10]. Using these experiments we can also measure something similar to **recall**, i.e. if the attribution method is identifying all of the contributing features. For instance, we can see from the insertion/deletion result shown in Fig. 5a that our proposed method has a lower deletion AUC. A lower deletion AUC means that more of the contributing features are removed compared to other methods, hence we can interpret it as having a higher recall. This is partly reflected in Fig. 2 as well. IBA, our method, and GradCAM are identifying ears and snout as contributing features. However, extremal perturbation is merely identifying the snout. The reason could be due to the formulation of the extremal perturbations method, which looks for the smallest mask that keeps the original output. Nevertheless, for drawing conclusions, we better resort to the deletion experiment introduced in Sec. 4.4.2.
>
> ---
>
> ##### **2) “Table 1: the results of IBA and integrated gradient are extremely bad. Is there a difference of scale of the attribution value which is causing IBA to highlight everything? For integrated gradient, what is the baseline? The confidence of the prediction score also have an effect on the input feature attribution. Did author look at them to make sure the they are running other methods with the optimal or reasonable settings?”**
>
> In the following, we discuss the results of IBA and Integrated Gradients separately:
>
> **IBA:** It is expected that IBA would perform poorly on IMDB dataset and the recurrent network setting. IBA relies on the assumption that there is a spatial correspondence between feature maps and the input (the correspondence exists in CNNs), such that IBA can map (scale) the feature mask back to input space as an attribution map. However, this assumption is basically not true in the RNN setting. As a result, we can see that the attribution score is roughly uniformly distributed across all words in one text. Increasing the hyperparameter $\beta$ in IBA won’t improve the result but just lower the attribution value uniformly for all words. The purpose of including IBA in this experiment is to demonstrate that it cannot be extended to this IMDB/RNN setting, while our solution can be.
>
> **Integrated Gradients:** In the case of Integrated Gradients for the qualitative example in Table 1, we normalized the attribution value to be in the [0,1] interval and applied the same visualization scheme for all methods as explained in the caption for Table 1. We truly appreciate the reviewer’s comment and agree that using the same visualization scheme may not be appropriate for this method in this qualitative example. In Fig. 5b we see that Integrated Gradients is performing well on the quantitative metric (Insertion/Deletion) on IMDB task and this is not reflected in the qualitative example in Table 1. The reason is that Integrated Gradients is assigning the order of importance of words correctly, however, the value is relatively very high for the most important ones. Thus the current normalization/visualization hurts the qualitative results for IG in Table 1, although our quantitative experiments in Sec. 4 introduce it on par with our own proposed method. We will change the normalization/visualization for IG to better reflect the results of insertion/deletion (Fig 5b), and add the discussion on visualization/normalization. Regarding the parameters of Integrated Gradients, we use the default (proposed in the original paper [2]) values. The number of integrated points is 50 and the baseline value is 0. We will add the details in the appendix. These values are commonly used and we observe that the method with default parameters performs well in the quantitative experiment (Fig. 5b) as explained.

---

### Decision · Program_Chairs · 2021-09-27

**Decision:**

Accept (Poster)

**Comment:**

The reviewers are all in consensus that the paper is worthy of acceptance.

I want to thank the authors for their extensive responses to the reviews.  It appears as though the reviews will benefit the paper and lead to improvements for the final form.